# 5-Methyl Furfural Reduces the Production of Malodors by Inhibiting Sodium l-Lactate Fermentation of *Staphylococcus epidermidis*: Implication for Deodorants Targeting the Fermenting Skin Microbiome

**DOI:** 10.3390/microorganisms7080239

**Published:** 2019-08-05

**Authors:** Manish Kumar, Binderiya Myagmardoloonjin, Sunita Keshari, Indira Putri Negari, Chun-Ming Huang

**Affiliations:** 1Department of Biomedical Sciences and Engineering, National Central University, Taoyuan 320009, Taiwan; 2Department of Life Sciences, National Central University, Taoyuan 320009, Taiwan; 3Department of Dermatology, School of Medicine, University of California, San Diego, CA 92093, USA

**Keywords:** deodorants, diacetyl, 5-methylfurfural, microbiome, *Staphylococcus epidermidis*

## Abstract

*Staphylococcus epidermidis* (*S. epidermidis*) is a common bacterial colonizer on the surface of human skin. *Lactate* is a natural constituent of skin. Here, we reveal that *S. epidermidis* used sodium l-lactate as a carbon source to undergo fermentation and yield malodors detected by gas colorimetric tubes. Several furan compounds such as furfural originating from the fermentation metabolites play a role in the negative feedback regulation of the fermentation process. The 5-methyl furfural (5MF), a furfural analog, was selected as an inhibitor of sodium l-lactate fermentation of *S. epidermidis* via inhibition of acetolactate synthase (ALS). *S. epidermidis* treated with 5MF lost its ability to produce malodors, demonstrating the feasibility of using 5MF as an ingredient in deodorants targeting malodor-causing bacteria in the skin microbiome.

## 1. Introduction

Studies have shown that body odors in humans are mainly derived by the bacterial decomposition of non-odorous constituents of sweats such as fatty acids, branched-chain aliphatic amino acids, glycerol, and lactic acid originated from apocrine, eccrine and sebaceous glands [1,2,3,4]. Human sweat mostly composed of *Staphylococcus* spp., *Cutibacterium* spp., and *Corynebacreium* spp., however there distribution varies in different age groups. The number of *Staphylococcus hominis* (*S. hominis*) is found to be significantly higher in children’s underarm than teenagers. However, *Staphylococcus epidermidis* (*S. epidermidis*) exhibits the strong underarm malodor in both age groups [5]. Diacetyl, or 2,3-butanedione, is a by-product of lactate, a major sweat constituent causing strong malodor and mostly produced by *Staphylococcus* spp., whereas no diacetyl production was detected from *Corynebacterium* spp. under experimental condition [6]. *Cutibacterium* spp. *such as Cutibacterium avidum* (*C. avidum*) predominantly causes malodor in underarm area, whereas, *S. epidermidis* is responsible for production of malodor in different parts of human skin [1,4,5,7,8].

Furthermore, enzymes in *S. epidermidis* are solely involved in multiple metabolic pathways resulting in generation of isovaleric acid and sour odor-associated acetic acid. Malodor results from degradation of branch chain amino acids such as leucine, valine, and isoleucine by acetolactate synthase (ALS) (EC: 2.2.1.6) and transaminase (EC: 2.6.1.42). These enzymes mostly involve in oxidative degradation of pyruvate aliphatic carboxylates to acetyl-CoA resulting in generation of acetic acid in *S. epidermidis* [5,9]. Unlike in *Cutibacterium striatum* (*C. striatum*)*,* a cystathionine beta-lyase enzyme was detected in *S. epidermidis*, which produces 3-methyl-3-sulfanylhexanol (3M3SH) in both children and teenager from a metagenomics study [5,10]. All above studies showed that acetic acid, isovaleric acid and 3M3SH in sweat are important odor precursors metabolized by the *Staphylococcus* spp. [5]. Diacetyl non-enzymatically produced from pyruvate by the process of oxidative decarboxylation. The threshold value of diacetyl is 100 times lower than the acetic acid [6]. ALS is found in plants and microorganisms. It can catalyse the conversion of pyruvate to α-acetolactate, which is subsequently broken down to *diacetyl* and acetoin [11]. A recent study has showed that diacetyl is as the major contributor to malodor inside the head region of middle-aged Japanese male and can cause an acid-like character to foot and axillary odor [12]. The concentration of lactate in skin is approximately 2.5 mmol/L which is higher than that (0.8 mmol/L) in the venous plasma [13]. However, a recent study detected the concentration of lactate in sweat before and after the exercise as 20.4 ± 6.7 and 62.2 ± 16.3 mmol/L respectively [14]. It has been illustrated that *Staphylococcus aureus* (*S. aureus*) and *S. epidermidis* in the human skin microbiome have the highest potential to convert lactate into diacetyl [6].

Nowadays, deodorants [15] consist of a trace amount of aluminum which has the potential to pass over the blood-brain barrier. In addition, the aluminum can accumulate inside hippocampus cortex’s pyramidal neurons, which may lead to the development of Alzheimer’s disease. In addition, free radicals created by aluminum may interfere with physiological processes involving magnesium and calcium [16]. Prostate and breast cancers may be associated with the use of aluminum antiperspirant [17]. When aluminum are accumulated in bone, it may lead to osteoporosis [18]. Commercially present deodorant consisting of hydrogenated castor oil (HCO) can lead to axillary dermatitis [19]. Triclosan (TCS) with antibacterial activity and methoxychlor (MXC), an endocrine-disrupting chemical (EDC) can be ingredients in deodorants. However, TCS may accelerate the growth of ovarian cancer cells through an estrogen receptor mediated signaling pathway [20]. MXC could induce endocrine-disrupting effects such as ovulation failures, uterine hypertrophy, atrophy of male sexual organs, and deteriorations of sperm production [21,22]. Transplantation of probiotic bacteria has been suggested to treat various skin problems including malodor production. Topical application of *Vitreoscilla filiformis* (*V. filiformis*) has been tested for treatment of atopic dermatitis. However, application of probiotic bacteria may exert a temporary effect and be not capable to permanently colonize on human skin [23]. Malodors derived from short chain fatty acids (SCFAs) can be produced through the bacterial fermentation. Besides SCFAs, furfural is one of by-products produced by bacterial fermentation. It has been documented that furfural displays the inhibitory effects on bacterial fermentation via blocking the activities of enzymes in the pathways of bacterial fermentation [24,25,26]. The lactate has been used as a carbon source for microbial fermentation to produce hydrogen [27]. Here, we use 5-methyl furfural (5MF), a furfural analog, to attenuate the production of malodors during the sodium l-lactate fermentation of *S. epidermidis*. 5MF effectively inhibits the activity of ALS in *S. epidermidis* and production of malodors. Selective targeting of odor-causing microbiome with 5MF may hold promise for treatment of underarm malodors.

## 2. Materials and Methods

### 2.1. Fermentation of Bacterial Culture and Fermentation

*S. epidermidis* (ATCC 12228) bacteria were cultured in 3% tryptic soy broth (TSB) (Sigma, St. Louis, MO, USA) overnight at 37 °C. The bacterial cultures were diluted 1:100 and cultured to an optical density 600 nm (OD_600_) = 1.0. Bacteria were collected by centrifugation at 5000 × *g* for 10 min and suspended in PBS for further experiments. For fermentation, *S. epidermidis* bacteria (10^3^ CFU/mL) with or without sodium l-lactate (0.02%) was incubated in 5 mL TSB media at 37 °C. The 0.002% (w/v) phenol red (Sigma), as a fermentation indicator, was added in the TSB at 37 °C for 12 h prior to pH measurement. A color change from red-orange to yellow detected by OD_562_ indicated the occurrence of bacterial fermentation.

### 2.2. Detection of Malodor Production

A device with a GT-92L tube (Gastec Corporation, Ayase-City, Kanagawa, Japan) which can detect the malodors including diacetyl, acetaldehyde and acetic acid, was used. In past, Gastec diffusion tubes were used for measurement of carbon monoxide (CO) as indicator for kitchen pollution [28]. The GT-92L tube was connected with a syringe before inserting to a sample tube containing *S. epidermidis* in TSB media with or without sodium l-lactate. After withdrawing the syringe, the evaporated malodors with a functional group of acetaldehyde were detected when chemical reagents such as ammonium phosphate in the GT-92L tube turned purple. Quantitative analysis of evaporated malodors was conducted using National Institutes of Health (NIH) ImageJ software by calculating the intensities of purple dots in a selected area (2.5 × 3 mm^2^) of a GT-92L tube. The intensities of purple dots in at least three GT-92L tubes per group were counted and expressed as the fold difference with respect to those in control.

### 2.3. Inhibition of ALS Activity by 5MF

ALS (EC: 4.1. 3.18) is in charge of the conversion of pyruvate to acetolactate which can be eventually metabolized to diacetyl and 2,3-butanediol. The rate of reaction was monitored by the depletion of Nicotinamide adenine dinucleotide (NADH) at 340 nm during the conversion of pyruvate to 2,3-butanediol. The reaction mixture contained 70 mM sodium acetate buffer (pH 5.4), 0.17 mM thiamine pyrophosphate, and lysate (10 mg) of heat (100 °C)-killed *S. epidermidis.* The reaction was started by addition of pyruvate [29] in the presence or absence of 0.15% 5MF at 45°C for 5 min. Absorbance was measured with a visible spectrophotometer 340 nm. Values of ALS activity were expressed in enzyme unit per mg (U/mg), in which one unit of ALS is defined as the amount of enzyme able to produce 0.1 absorbance unit per min.

### 2.4. Inhibitory Effect of 5MF on Fermentation and Malodor Production

*S. epidermidis* (10^3^ CFU/mL) and sodium l-lactate (0.02%) in phenol red-containing TSB media with or without 0.15% 5MF were incubated 12 h at 37 °C. The effect of 5MF on bacterial fermentation was monitoring by reading OD_562_. To determine the effect of 5MF on malodor production, *S. epidermidis* (10^3^ CFU/mL) was incubated with or without 0.15% 5MF for 48 h before adding into a sample tube containing TSB media with 0.02% sodium l-lactate. The sample tube was subsequently connected to a GT-92L tube for detection of malodors as shown in Section 2.2.

### 2.5. Cell Viability Assay

The CCD 1106 KERTr (ATCC CRL-2309), a human keratinocytes cell line, was seeded at a density of 5 × 10^3^ cells/well in 96 well plates. Cells were treated with or without 0.15%, 0.17%, or 0.20% 5MF for 24 h. 5 mg/well of 3,4,5-Dimethyl thiazol-2-yl)-2-5 diphenyl tetrazolium bromide (MTT) (Thermo Scientific, Waltham, MA, USA) was added into each well and incubated at 37 °C, 5% CO_2_ for 2 h. After incubation, the media were removed and 200 μL/well of dimethyl sulfoxide (DMSO) (Sigma) was added to dissolve the formazan crystals. The solubilized crystals were then quantified by scanning the plates at 570 nm using a Biochrom EZ Read 400 Microplate Reader (Biochrom Ltd., Cambourne, Cambridge, UK). The cell viability was expressed as a percent using the following formula: cell viability (%) = [A/A_0_] × 100, where A and A_0_ are the absorbance of mean values of cells treated with or without 5MF, respectively.

### 2.6. Statistical Analysis

All experiments were performed in triplicate. Data analysis was performed by unpaired *t*-test where the *p*-values of < 0.05 (*), < 0.01 (**), and < 0.001 (***) were accepted for statistical significance.

## 3. Results

### 3.1. S. epidermidis Ferments Sodium l-Lactate

To examine whether *S*. *epidermidis* (ATCC 12228) bacteria can ferment sodium l-lactate, bacteria were incubated with and without sodium l-lactate (0.02%) in the TSB media with phenol red for 12 h. TSB with and without sodium l-lactate in the absence of bacteria served as controls. Phenol red was used as a fermentation indicator to monitor the bacterial fermentation. The color of phenol red in TSB media incubated with bacteria changed to orange due to the bacterial replication during incubation. TSB media with bacteria plus sodium l-lactate turned yellow after 12 h of incubation, indicating the sodium l-lactate fermentation of *S. epidermidis* has occurred (Figure 1A). The color change of phenol red was quantified by measuring the OD at 562 nm (OD_562_). The value of OD_562_ in TSB media with sodium l-lactate and *S. epidermidis* was significantly lower than that in TSB media with *S. epidermidis* alone (Figure 1B).

### 3.2. Detection of Malodors by A Gas Colorimetric Tube

A gas colorimetric tube (GT-92L) for measurement of malodors including was used for detection of malodors produced by sodium l-lactate fermentation of *S. epidermidis*. As shown in Figure 2A, a device with a 25 mL syringe connected with a GT-92L tube was inserted into a sample tube which contained 200 μL 97% diacetyl. After syringe suction, the color of chemical reagents in the gas colorimetric tube turned purple, demonstrating the effectiveness of the device for detection of malodors. The device was inserted to samples tubes containing TSB media without phenol red (tube B1), TSB media plus sodium l-lactate (tube B2), TSB media plus *S. epidermidis* (tube B3) or TSB media plus *S. epidermidis* and sodium l-lactate (tube B4). Malodors were not detectable in tube B1 or B2 which was filled with TSB media or TSB media plus sodium l-lactate. Little malodors appeared in tube B3 which had *S. epidermidis* in TSB media. However, the high level of malodor was detected in tube 4 which contained TSB media with *S. epidermidis* plus sodium l-lactate (Figure 2B). The results illustrated the malodors including diacetyl were generated during the sodium l-lactate fermentation of *S. epidermidis*. Quantitative analysis indicated that the levels of malodors in tube B4 were approximately two times higher than those in tube B3 (Figure 2C).

### 3.3. Inhibition of ALS Activity by 5MF

In the pathway of bacterial fermentation, ALS with approximately 60 kDa in size is involved in converting pyruvate to 2-acetolactate which can be subsequently decomposed to diacetyl or be decarboxylated to acetoin [30]. Acetoin can be metabolized to 2,3-butanediol in the presence of NADH. Inhibition of ALS activity will block the production of diacetyl and 2,3-butanediol. Microbial fermentation generates several by-products including furfural and its analogue, 5-hydroxymethylfurfural (5-HMF), which exert the inhibitory effects on microbial fermentation via interfering with the activities of enzymes in the pathways of microbial fermentation [24,25]. Although 5MF, a furfur analog, at the concentrations of 0.15%, 0.17% and 0.20% was not toxic to skin keratinocytes, the 0.17% and 0.20% 5MF suppressed the growth of *S. epidermidis* (Appendix A). Thus, 0.15% 5MF was chosen to examine if its effect on the activity of ALS. As shown in Figure 3, 0.15% 5MF effectively reduced the activities of ALS.

### 3.4. The Influence of 5MF on Sodium l-Lactate Fermentation of S. epidermidis

To investigate if 5MF with the inhibitory activity of ALS influences the sodium l-lactate fermentation of *S. epidermidis*, 0.15% 5MF was added into the TSB media containing *S. epidermidis* (ATCC 12228) and 0.02% sodium l-lactate. In agreement with the results in Figure 1, incubation of *S. epidermidis* and sodium l-lactate in phenol red-containing TSB media for 12 h turned media yellow and resulted in a significant decrease in the value of OD_562_, indicating the occurrence of the sodium l-lactate fermentation of *S. epidermidis.* Red media and a higher value of OD_562_ were detected when 5MF was added into media containing *S. epidermidis* and sodium l-lactate (Figure 4). The pH value of 0.15% 5MF is 7.1 which excluded the possibility of lowering a pH value of medium by addition of 0.15% 5MF (data not shown). Taken together, results in Figure 3 and Figure 4 suggested that 5MF hindered the sodium l-lactate fermentation of *S. epidermidis* by inhibition of ALS activity.

### 3.5. Reduction of Malodor Production by 5MF

To assess whether inhibition of ASL activity and sodium l-lactate fermentation of *S. epidermidis* by 5MF can lead to a reduction of the production of malodors during bacterial fermentation, *S. epidermidis* bacteria were pre-treated with or without 0.15% 5MF for 48 h before transferring to sample tubes of the culture media with 0.02% sodium l-lactate. The device with a GT-92L tube connected with a syringe (Figure 2) was subsequently inserted into the sample tubes to monitor the production of malodors for 5 min. As shown in Figure 5A, *S. epidermidis* without 5MF pre-treatment can effectively generate malodors (purple appearing on a GT-92L tube) via sodium l-lactate fermentation. However, when bacteria were pre-treated with 5MF, the production of malodors was reduced by 20% (Figure 5B), demonstrating the efficacy of 5MF as an inhibitor for reducing malodors produced by sodium l-lactate fermentation of *S. epidermidis*.

## 4. Discussion

Regional variations in sweat composition have been reported [31]. Human sweat contained approximately 0.1% lactate [32]. In this study, we utilized the 0.02% sodium l-lactate for *S. epidermidis* fermentation (Figure 1) since the higher sodium l-lactate may exert a toxic effect on the bacterial growth [33]. Lactate is fermented to acetate and propionate by various bacteria including *Propionibacterium spp.* via the methylmalonyl-CoA or acrylyl-CoA pathway [34]. It has been documented that *Megasphaera elsdenii* (*M. elsdenii*) can use the lactate as a carbon source for fermentation and produce hydrogen and other metabolites including acetate and propionate [27]. Moreover, presence of butyrate in isolates obtained from 10^-8^ dilutions of fecal samples from five different subjects depicts that bacteria might have utilized the lactate to produce butyrate [35]. Although we have detected malodor compounds produced by sodium-l-lactate fermentation of *S. epidermidis* (Figure 2), these compounds will be identified by high-performance liquid chromatography (HPLC) and/or gas chromatography mass spectrometry (GC-MS).

Both furfural and 5-hydroxy methyl furfural (5-HMF) have furan rings composed of C_5_H_4_O_2_ and C_6_H_6_O_3_, respectively. Furfural is one of by-products of microorganism fermentation. It, however, exerts the inhibitory effects on the process of fermentation for the conversion of carbon sources to the end production [36,37]. Furfural and HMF have been reported to have inhibitory effects on the specific growth rate, as well as fermentation rate of microorganism [38]. Since furan derivatives can be naturally produced by fermentation of microorganisms, we here utilized 5MF, a furfural analog, as an inhibitor of sodium l-lactate fermentation of *S. epidermidis* (Figure 4). 5MF has been found a natural ingredient in vinegars [39] and is using as a flavouring agent in a variety of *food* products. Blockade of fermentation via down-regulation of ALS activities (Figure 3) by 5MF efficiently reduced the production of malodors (Figure 5), demonstrating the feasibility of using 5MF as an ingredient for development of deodorants to treat body malodors.

Results by kinetic analysis revealed the function of furfural as a fermentation inhibitor is mediated by competitive inhibition for alcohol dehydrogenase (ADH) and aldehyde dehydrogenase (AlDH) and non-competitive inhibition for pyruvate dehydrogenase (PDH) [40]. ALS catalyzes the first common step in branched chained amino acid synthesis in plants. Several classes of commercially used herbicides, including sulfonylureas, imidazolinones, and sulfonanilides, target this pathway through the inhibition of ALS [41,42]. The chemogenomic profile revealed both the regulatory (Ilv6p) and catalytic (Ilv2p) subunits of ALS in fungi [43]. It has been confirmed that catalytic subunit of ALS is the primary binding target for the triazolopyrimidine series of compounds with broad-spectrum antifungal activity [44]. Although our data demonstrated the inhibitory effect of 5MF on ALS, future studies will determine how 5MF subdues the activity of ALS.

## 5. Conclusions

Lactate in human skin may be a natural carbon source to induce the fermentation of bacteria in the skin microbiome. Lactate fermentation of skin bacteria could be a cause of body malodors. Here, we demonstrate that *S. epidermidis*, a bacterial member in the human skin microbiome, fermented sodium l-lactate and produced detectable malodors. The malodor production from sodium l-lactate fermentation of *S. epidermidis* was effectively suppressed by inhibition of ALS using 5MF, an analog of furfural which has previously been known to negatively regulate the fermentation process.

## Figures and Tables

**Figure 1 microorganisms-07-00239-f001:**
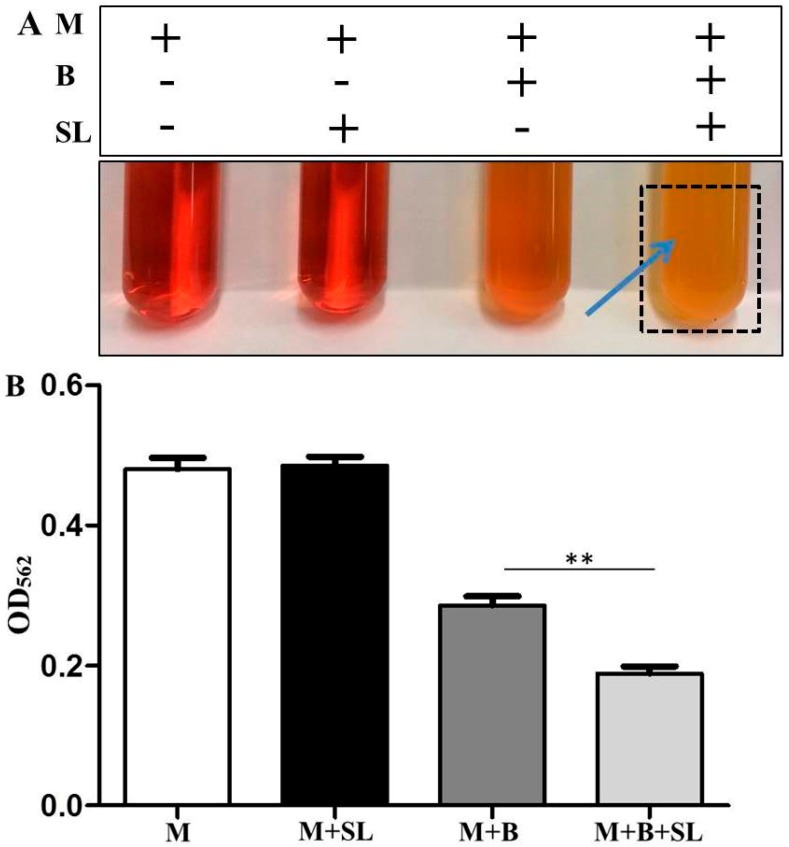
Sodium l-lactate fermentation of *S. epidermidis*. (**A**) *S. epidermidis* bacteria (B) (10^3^ CFU/mL) were incubated in tryptic soy broth (TSB) media (M) containing phenol red with or without 0.02% sodium l-lactate (SL) for 12 h. TSB media alone or media with sodium l-lactate were included as controls. Bacterial fermentation was indicated by the color change of phenol red to yellow (arrow). (**B**) A graph showing the OD_562_ value in media with bacteria plus sodium l-lactate (M+B+SL) was significantly lower than that in media with bacteria (M+B). Results were illustrated as mean ± standard deviation (SD) of three independent experiments. ** *p* < 0.01 (two-tailed *t-test*).

**Figure 2 microorganisms-07-00239-f002:**
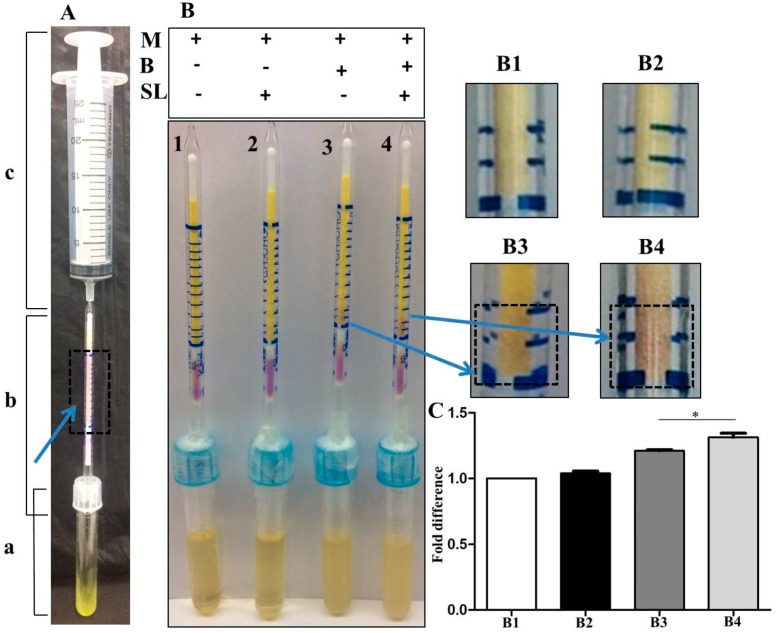
The detection of malodors. A GT-92L colorimetric tube was used for the detection of malodors produced by sodium l-lactate (SL) fermentation of *S. epidermidis* (B). (**A**) A device with a GT-92L tube (b) connected with a syringe (c) was inserted to a sample tube (a) containing 200 μL 97% diacetyl. Evaporated diacetyl was detected when chemical reagents in the GT-92L tube turned purple (arrow). (**B**) The devices were inserted into four samples tubes containing TSB media (M) only, media with bacteria (M+B), media with sodium l-lactate (M+SL) and media with bacteria and sodium l-lactate (M+B+SL). No phenol red was added into the media. The zoomed panels (B1–B4) from four tubes indicated that, 5 min after device insertion, evaporated malodors (purple) were abundantly detected in a sample tube containing media with bacteria and sodium l-lactate. The representative data was taken from three independent experiments. (**C**) The levels of malodors in each sample tube were quantified and expressed as the fold difference relative to the level of malodor in sample tube B1. Results were shown as mean ± SD of three independent experiments. * *p* < 0.05 (two-tailed *t*-test).

**Figure 3 microorganisms-07-00239-f003:**
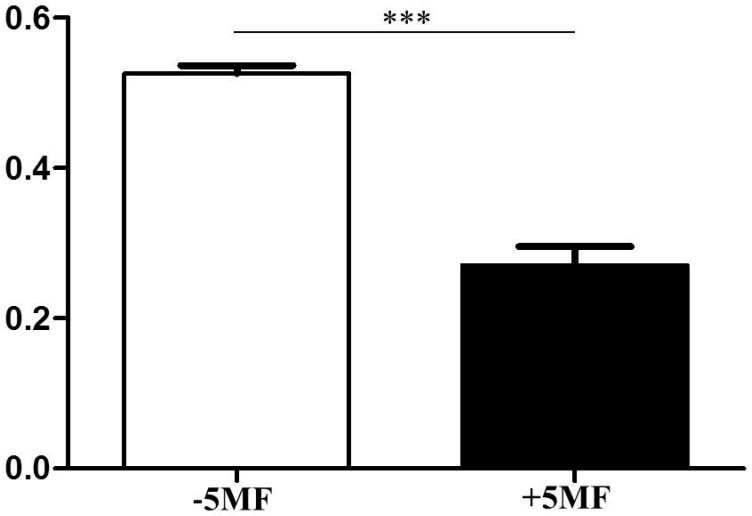
Inhibition of acetolactate synthase (ALS) activity by 5MF. The reaction mixture containing lysates of *S. epidermidis* was incubated with (+ 5MF) or without 5MF (− 5MF). The activity (U/mg) of ALS in *S. epidermidis* lysates was quantified. Data shown represent the mean ± SD of experiments performed in triplicate. *** *p* < 0.001 (two-tailed *t*-test).

**Figure 4 microorganisms-07-00239-f004:**
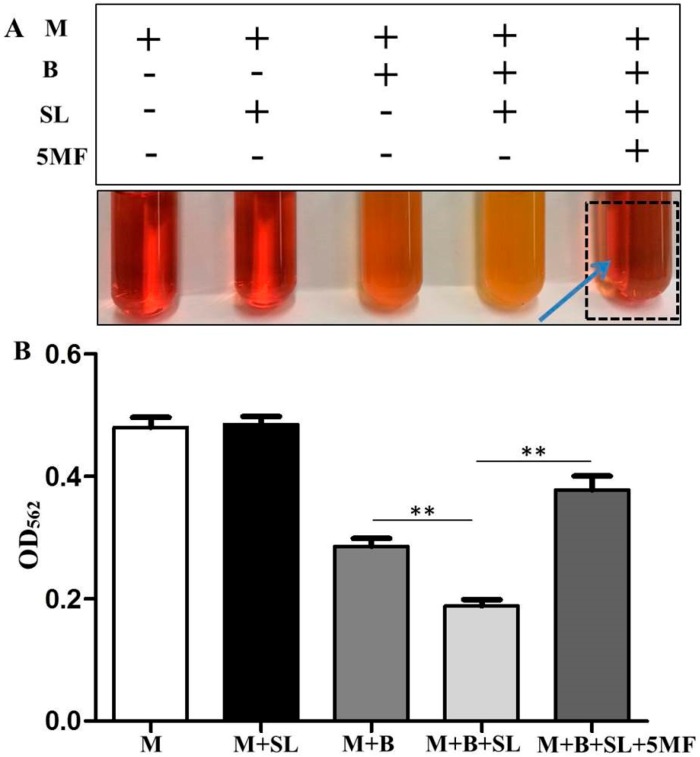
Suppression of sodium l-lactate fermentation of *S. epidermidis* by 5MF. (**A**) *S. epidermidis* (B) (10^3^ CFU/mL) was cultured in a tube containing phenol red, TSB media and 0.02% sodium l-lactate (SL) in the presence (M+B+SL+5MF) or absence (M+B+SL) of 0.15% 5MF for 12 h. Tubes containing TSB media (M), media plus sodium l-lactate alone (M+SL) or bacteria alone (M+B) were included as controls. (**B**) The values of OD_562_ all tubes were quantified. Addition of 5MF restored a decrease in the value of OD_562_ caused by sodium l-lactate fermentation of *S. epidermidis*. The red appearance (arrow) of culture media containing bacteria, sodium l-lactate and 5MF was observed. Data are the mean ± SD of three separate experiments. ** *p* < 0.01 (two-tailed *t*-tests).

**Figure 5 microorganisms-07-00239-f005:**
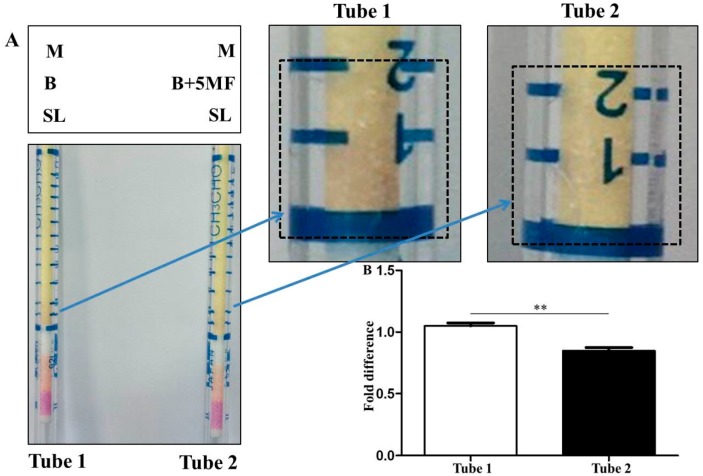
Reduction of malodor production by 5MF. (**A**) Two samples tubes containing TSB media (M), sodium l-lactate (SL), and *S. epidermidis* pre-treated with (Tube 2) or without (Tube 1) 0.15% 5MF for 48 h. The sample tubes were connected to GT-92L colorimetric tubes. Zoomed panels indicated the change in the production of evaporated malodors (purples in squares) 5 min after connection of sample tubes to GT-92L calorimetric tubes. The representative images of three similar results were shown. (**B**) The levels of malodors were quantified and expressed as the fold difference relative to those in Tube 1. Results were shown as mean ± SD of three separate experiments. ** *p* < 0.01 (two-tailed *t*-test).

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
