# Peer review of "5-Methyl Furfural Reduces the Production of Malodors by Inhibiting Sodium l-Lactate Fermentation of Staphylococcus epidermidis: Implication for Deodorants Targeting the Fermenting Skin Microbiome"

_microorganisms, 2019, doi:10.3390/microorganisms7080239_

Round 1

Reviewer 1 Report

The authors of the manuscript entitled "5-methyl Furfural Reduces the Production of Malodors by Inhibiting Sodium L-lactate Fermentation of Staphylococcus epidermidis: Implication for Deodorants Targeting the Fermenting Skin Microbiome" have presented a clear manuscript demonstrating the effect of their molecule against the fermentation of malodors.  The manuscript was almost acceptable as presented, but a few minor details are neccessary to fix. 

Page 2 line 61: "may I interfere"  -- remove I.

Page 3 line 90: change was to were, bacteria were

Page 3 line 114: MF -- is this 5MF?

Page 4 line 136: A0 should have a subscript 0

Page 3 line 104: What % is the area sampled compared to the overall sample size?  How representative/subjective can this selection be?

Page 5 Figure 1: The standard deviations are quite small, are the lines presented confirmed to be standard deviations, or are they standard errors?

Page 6 lline 181: The 2-fold difference in the text and refered to Fig 2C is not obvious as presented.  You have a fold difference of 1.0 as standard, then 1.2 for tube B3 and 1.3 for B4.  Thus the 2-fold difference is not obvious based on the presentation in Fig. 2C (This works if 0.1 fold of B1 is greater than or equal to raw B3).

Page 8 Figure 4 caption:  The two instances of the use of (B) is confusing, despite there being a bold (B) indicating panel B.  Please use a different terminology for describing the bacteria in panel A.

Figure 1B and 4B appear to be the same figure, only with 4B having an additional 5MF bar.  In these figures, the medium without sodium L-lactate is showing the same fermatation as the medium supplemented with sodium L-lactate.  How is this possible?  I understand that the reduction of wavelength 562 indicates fermentation of lacate, but then shouldn't additional lactate have an increased absorbance at 562 nm?

Page 9 Figure 5: How does tube 1 have a 0 SD?  Since this is normalized, the data should be taken not as 1.0 for each tube one separately, but take the raw average of tube 1 and make that 1.0.  Then Tube 1 will have an average of tube 1, but the three values may differ with respect to the average, providing evidence to the amount of error in the measurement.

Author Response

Responses to Reviewers 

We appreciate the thoughtful and insightful comments made by the reviewers regarding our manuscript (microorganisms-536780) entitled “5-methyl Furfural Reduces the Production of Malodors by Inhibiting Sodium L-lactate Fermentation of Staphylococcus epidermidis: Implication for Deodorants Targeting the Fermenting Skin Microbiome “. Changes are underlined in the text of manuscript. 

Reviewer 1 

The authors of the manuscript entitled "5-methyl Furfural Reduces the Production of Malodors by Inhibiting Sodium L-lactate Fermentation of Staphylococcus epidermidis: Implication for Deodorants Targeting the Fermenting Skin Microbiome" have presented a clear manuscript demonstrating the effect of their molecule against the fermentation of malodors. The manuscript was almost acceptable as presented, but a few minor details are necessary to fix. 

Comment 1 

Page 2 line 64: "may I interfere" -- remove I. 

Response 1 

We have corrected it. 

Comment 2 

Page 2 line 89: change was to were, bacteria were 

Response 2 

We have corrected it. 

Comment 3 

Page 3 line 114: MF -- is this 5MF? 

Response 3 

Answer: Yes, in page 3 line 113 is 5MF and we have corrected it. 

Comment 4 

Page 3 line 133: A0 should have a subscript 0. 

Response 4 

Answer: We have corrected it. 

Comment 5 

Page 3 line 104: What % is the area sampled compared to the overall sample size? How 

representative / subjective can this selection be? 

Response 5 

Answer: The percentages of sodium lactate and 5MF are 0.02% and 0.15%, respectively. The bacterial number is 103 CFU/mL in a 5 mL of media. From Figures 1 to 5, all are showing similar results with this overall sample size. 

Comment 6 

Page 5 Figure 1: The standard deviations are quite small, are the lines presented confirmed to be standard deviations, or are they standard errors? 

Response 6 

Page 5 Figure 1B, lines presented are standard deviations. 

The experiment in Figure 1 has been repeated for three times. Figure 1 is replotted with new data values with a standard deviation. 

Comment 7 

Page 6 line 181: The 2-fold difference in the text and referred to Fig 2C is not obvious as presented. You have a fold difference of 1.0 as standard, then 1.2 for tube B3 and 1.3 for B4. Thus the 2-fold difference is not obvious based on the presentation in Fig. 2C (This works if 0.1 fold of B1 is greater than or equal to raw B3). 

Response 7 

In page 6 line 181, in Figure 2C, the term of 2-fold difference in the text was used by measurement of densitometry analysis of color intensity by NIH Image J software.  The particular area of color intensity in B4 was detected two-times higher than in B3. We have replaced the submitted image data in Figure 2C with a high-resolution image. Further densitometry analysis for color intensity has been done and graph is modified. 

Comment 8 

Page 7 Figure 4 caption: The two instances of the use of (B) is confusing, despite there being a bold (B) indicating panel B. Please use a different terminology for describing the bacteria in panel A. 

Response 8 

Page 7 Figure 4, the first instance we used B for panel B. Inside the Figure we used B for indicating bacteria as this terminology we keeping constant for our all microbiology work. We kindly request the reviewer to allow us use the same terminology for bacteria.  

Comment 9 

Figure 1B and 4B appear to be the same figure, only with 4B having an additional 5MF bar. In these figures, the medium without sodium L-lactate is showing the same fermentation as the medium supplemented with sodium L-lactate. How is this possible? I understand that the reduction of wavelength 562 indicates fermentation of lactate, but then shouldn't additional lactate have an increased absorbance at 562 nm? 

Response 9 

Yes, we have done same experiment for Figures 1 and 4 with and without inhibitor. Now, we have repeated the experiment for Figures 1 and 4. We are presenting the new graph and OD at 562 nm graph in Figures 1B and 4B.  

In Figure 1B and 4B, the media without sodium L-lactate contain bacteria. Here we have used TSB (Tryptic soy broth) media which contain few carbon metabolites that might have used by bacteria as a carbon source during bacterial growth, turning phenol red color from red to yellowish-orange. However, in the media with sodium Llactate and bacteria, we have specifically used sodium L-lactate as a carbon source to detect its fermentation activity. We have detected significantly higher fermentation with L-lactate changing the color of phenol red from red to yellow. We have also cited two references in our first submission (no. 27 and no 35), evidencing lactate is a carbon source for fermentation. 

[27] Ohnishi, A.; Hasegawa, Y.; Abe, S.; Bando, Y.; Fujimoto, N.; Suzuki, M. Hydrogen fermentation using lactate as the sole carbon source: Solution for ‘blind spots’ in biofuel production. RSC Advances 2012, 2, 8332-8340, doi:10.1039/C2RA20590D. 

[35] Duncan, S.H.; Louis, P.; Flint, H.J. Lactate-utilizing bacteria, isolated from human feces, that produce butyrate as a major fermentation product. Appl Environ Microbiol 2004, 70, 5810-5817, doi:10.1128/AEM.70.10.5810-5817.2004.   

Comment 10 

Page 8 Figure 5: How does tube 1 have a 0 SD? Since this is normalized, the data should be taken not as 1.0 for each tube one separately, but take the raw average of tube 1 and make that 1.0. Then Tube 1 will have an average of tube 1, but the three values may differ with respect to the average, providing evidence to the amount of error in the measurement. 

Response 10 

In Page 8 Figure 5, in tube 1 pretreated with TSB media, sodium L-lactate, and S. epidermidis, we have detected presence of diacetyl and we have used it as a reference to measure the diacetyl production in tube 2 pretreated with TSB media, sodium Llactate, S. epidermidis and 5MF. 

Reviewer 2 Report

The study demonstrated by Kumar et al. is very interesting and well written to clarify the 5MF reduce the production of Molodors by inhibiting sodium L-lactate fermentation of Staphylococcus epidermidis.

The major concerns for the reviewer are the following.

In Fig. 2; 

The authors explained as follows; the zoomed panels (B1-B4) from four tubes indicated that, 5 min after device insertion, evaporated malodors (purple) were abundantly detected. However, the reviewer cannot detect the purple color clearly. The photograph should be clear to see the purple color. Furthermore, the reviewer cannot understand how accumulate the level of malodor exactly. For example, Is accumulation of malodor measured by the length of purple color-changing in the column? 

In Fig. 1, the authors indicated that medium (TSB) plus bacteria induced L-lactate fermentation, and adding L-lactate significantly enhanced the fermentation. The reviewer cannot understand whether the assay is L-lactate specific fermentation. If it is true, why the L-lactate specific fermentation was detected in M+B? Please explain clearly.

In Fig. 4, the reviewer can see that turbidity in tube of M+B+SL+5MF is lesser than that of M+B+SL. Is 5MF toxic for bacteria? Have the authors confirmed whether the toxicity of 5MF for bacteria? 

Author Response

Responses to Reviewers 

We appreciate the thoughtful and insightful comments made by the reviewers regarding our manuscript (microorganisms-536780) entitled “5-methyl Furfural Reduces the Production of Malodors by Inhibiting Sodium L-lactate Fermentation of Staphylococcus epidermidis: Implication for Deodorants Targeting the Fermenting Skin Microbiome “. Changes are underlined in the text of manuscript.  

Reviewer 2 The study demonstrated by Kumar et al. is very interesting and well written to clarify the 5MF reduce the production of Malodors by inhibiting sodium L-lactate fermentation of Staphylococcus epidermidis

Comment 1 

The authors explained as follows; the zoomed panels (B1-B4) from four tubes indicated that, 5 min after device insertion, evaporated malodors (purple) were abundantly detected. However, the reviewer cannot detect the purple color clearly. The photograph should be clear to see the purple color. Furthermore, the reviewer cannot understand how accumulate the level of malodor exactly. For example, is accumulation of malodor measured by the length of purple color-changing in the column? 

Response 1

The appearance of purple is in the case of pure diacetyl which is shown in Figure 2A. In the zoomed panels, especially B3 and B4 are the mixture product of various malodors. For example, diacetyl, acetic acid, propionic acid and etc. These malodor gases are detected by the specialized tubes known as GASTEC detector 92L tubes. We have replaced the previous image with a high-resolution image for better visibility of diacetyl detection.   Before the insertion of GASTEC tubes, the samples were put in boiled water (900C), as we know that the boiling point of diacetyl is 880C. Most of malodors gases might have generated in the chamber and then we detected the gases by inserting the tubes.   

Comment 2

 In Fig. 1, the authors indicated that medium (TSB) plus bacteria induced Llactate fermentation, and adding L-lactate significantly enhanced the fermentation. The reviewer cannot understand whether the assay is L-lactate specific fermentation. If it is true, why the L-lactate specific fermentation was detected in M+B? Please explain clearly. 

Response 2 

Also see Response 9. In Figure 1, the media without sodium L-lactate contain bacteria. Here we have used TSB (Tryptic soy broth) media which contain few carbon metabolites that might have used by bacteria as a carbon source during bacteria growth turning phenol red color from red to yellowish-orange. However, in the medium with sodium L-lactate and bacteria, we have specifically used sodium L-lactate as a carbon source to detect its fermentation activity and we have detected significantly higher fermentation with L-lactate changing the color of phenol red from red to yellow. We have also cited two references in our first submission (no. 27 and no 35), evidencing lactate is a carbon source for fermentation. The main purpose to use lactate as a carbon source for fermentation is for the production of diacetyl. We have also cited a reference (no. 6) in our first submission evidencing lactate as a major precursor diacetyl production. 

[27] Ohnishi, A.; Hasegawa, Y.; Abe, S.; Bando, Y.; Fujimoto, N.; Suzuki, M. Hydrogen fermentation using lactate as the sole carbon source: Solution for ‘blind spots’ in biofuel production. RSC Advances 2012, 2, 8332-8340, doi:10.1039/C2RA20590D.

 [35] Duncan, S.H.; Louis, P.; Flint, H.J. Lactate-utilizing bacteria, isolated from human feces, that produce butyrate as a major fermentation product. Appl Environ Microbiol 2004, 70, 5810-5817, doi:10.1128/AEM.70.10.58105817.2004.  

 [6] Hara T, Matsui H, Shimizu H. Suppression of microbial metabolic pathways inhibits the generation of the human body odor component diacetyl by Staphylococcus spp. PLOS ONE 2014, 9, e111833, doi: 10.137/journal.pone.0111833. 

Comment 3

 In Fig. 4, the reviewer can see that turbidity in tube of M+B+SL+5MF is lesser than that of M+B+SL. Is 5MF toxic for bacteria? Have the authors confirmed whether the toxicity of 5MF for bacteria? 

Response 3 

With MBC (Minimum bactericidal concentration assay) assay, we confirm that 5MF is not toxic to bacteria and data is present in the “Supplementary Information”. 

Round 2

Reviewer 2 Report

The reviewer understood the authors' responses and recognized that the revised version is suitable to accept.